# Maternal information-seeking on pregnancy-induced hypertension and associated factors among pregnant women, in low resource country, A cross-sectional study design

**Ayana Alebachew Muluneh** [1]*, **Fekade Demeke Bayou**[1], **Kegnie Shitu**[2], **Ayenew Sisay Gebeyew**[3], **Sefefe Birhanu Tizie**[3], **Mulugeta Desalegn Kasaye**[1], **Adamu Ambachew Shibabaw**[4], **Agmasie Damtew Walle**[4]

**1** Department of Health Informatics, School of Public Health, College of Medicine and Health Science, Wollo University, Dessie, Ethiopia, **2** Department of Health Education and Behavioral Sciences, Institute of Public Health, University of Gondar, Gondar, Ethiopia, **3** Department of Health Informatics, College of Medicine and Health Sciences, Debre Markos University, Debre Markos, Ethiopia, **4** Departments of Health Informatics, College of Health Science, Mettu University, Mettu, Ethiopia

* alayu382@gmail.com

## Abstract

Pregnancy-induced hypertension is the most prevalent medical problem associated with pregnancy. It has been reported to affect 6–10% of all pregnant women worldwide. Mothers' failure to seek information related to PIH increases the risk of death from the complication of pregnancy-induced hypertension. This study aimed to assess PIH information-seeking behaviour and its associated factors among pregnant women in rural Sekela Woreda. A community-based cross-sectional study was conducted from May 15 to June 15, 2022. An interviewer-administered structured questionnaire was used to collect the data. The sample size was 635. A cluster sampling technique was used to select the sampled kebeles. The study population included rural pregnant women. This study included pregnant women who were permanent residents of the study area, whereas this study excluded pregnant women who were admitted only for delivery services and temporary residents who visited the study area. The mean age of the participants was 31.8 ± 6.09 years, with minimum and maximum ages of 20 and 45 years, respectively. We conducted descriptive analysis, bivariable analysis, and multivariable analysis to identify determinants of PIH information seeking. The proportion of pregnancy-induced hypertension (PIH) information seeking among pregnant women was 214 (35.4%) out of 604. Pregnant mothers aged 35 years and above (AOR =0.67, 95% CI =0.46, 0.97), family resistance (AOR = 0.45, 95% CI =0.29, 0.69), health care satisfaction (AOR =1.7, 95% CI =1.1, 2.5), and perceived severity of pregnancy-induced hypertension (PIH) (AOR =1.6, 95% CI =1.1, 2.4) were significantly associated with pregnancy-induced hypertension information seeking. According to our findings Information seeking related to pregnancy-induced hypertension is low. Aged mothers, family resistance, mothers' satisfaction with health care services, and perceived severity of PIH were found to be associated with pregnancy-induced hypertension information seeking. Expanding health education programs for pregnant women and providing awareness and training about PIH

**Data availability statement:** Data is available as supporting information file.

**Funding:** The author(s) received no specific funding for this work.

**Competing interests:** The author has declared that no competing interests exist.

to participants and their husbands is the most effective way to reduce the prevalence of PIH complications.

## Author summary

Pregnancy-induced hypertension is a unique kind of hypertension that happens at 20 weeks or more in pregnant individuals with or without proteinuria and comprises gestational hypertension, preeclampsia, and eclampsia. The most common pregnancy-related medical issue, approximately 5–10% of pregnant women, or 8 billion pregnant women, including starts, are affected; it is a cause of maternal mortality and mortality worldwide. Pregnant women who delay getting information on pregnancy-induced hypertension are more likely to die from problems related to Pregnancy-induced hypertension. Pregnant women ought to get adequate information such as miscarriage, pregnancy complications/danger signs, pregnant medications, diet, and the place of delivery. However, there is a limitation of research in limited resource settings regarding maternal information seeking on pregnancy-induced hypertension. The study found that a total of 65.47% of participants sought information once a month and 3.39% sought information daily about pregnancy-induced hypertension. Therefore, this research has a greater implication for enhancing the magnitude of maternal information seeking regarding pregnancy-induced hypertension. Finding the best source of information on pregnancy-induced hypertension complications aids in knowing where to go when pregnancy-induced hypertension complications happen, and it lowers the prevalence of its complications by lowering family resistance and increasing the severity of the case.

## Introduction

Pregnancy-induced hypertension (PIH) is a new form of hypertension that occurs at 20 weeks or more in pregnant patients with or without proteinuria and includes gestational hypertension, preeclampsia, and eclampsia [1,2]. Hypertensive disorders of pregnancy are the major complications that cause approximately 60–80% of all maternal deaths [3]. PIH is a significant public health warning and cause of neonatal morbidity and mortality, especially in developing countries, accounting for more than 10 million low birth weight infants, and women who have PIH are nearly ten times more at increased risk of low birth weight [4].

   PIH information-seeking behaviour is regularly vital for the advancement of knowledge, behavioural changes, health decisions, and the sharing of PIH health information among women [5]. Currently, health information-seeking behaviour (HISB) is broadly viewed as how individuals obtain information about health, illness, health promotion, and risks [6]. Information-seeking behaviours is a form of self-fulfilment involving the gathering of information about health from a variety of sources [7]. Information on health that leads to psychosocial adjustment of the known risk is crucial for health promotion [8]. Health information is used for the prevention, control, and psychological arrangement of individuals for the occurrence of risk or disease, and this is not only for the occurrence of risk or disease but also for life management [9]. PIH is a cause of maternal morbidity and mortality globally; it affects approximately 5–10%, or 8 billion pregnant women, including starters [10,11]. During pregnancy, there is so much information that women need to be fed. Women should be adequately educated on choosing the place of delivery, the pregnancy period, nutrition, miscarriage, pregnancy complications/danger signs, and medicines during pregnancy [12].

When pregnant mothers delay seeking PIH information, their risk of dying due to PIH complications increases [13]. Information-seeking behaviour differs from person to person in terms of the method used by the information seeker to access the information through their thoughts, emotions, and actions [14].

In many developing nations, low health literacy is a public health issue [15]. Low maternal health literacy is one of the various leading causes, which also include social and demographic problems, incomplete accessibility, and poor maternal and child health service quality [16]. Health literacy may influence an individual's information seeking, as well as her health understanding and actions. A study revealed that pregnant women's health literacy and information seeking are positively connected [15].

Information seeking is a process in which a person purposefully needs to change the status of their knowledge. People use this information to solve problems, do work, or increase their knowledge [3]. Worldwide, 10% of pregnant women are complicated by hypertension, with preeclampsia and eclampsia being the major causes of maternal and perinatal morbidity and mortality [11]. Pregnancy-induced hypertension is the most prevalent medical problem associated with pregnancy, affecting 6–10% of all pregnancies worldwide [17,18].

A study conducted in the Tigray region revealed that the prevalence of pregnancy-induced hypertension in Ethiopia ranges from 2.2% to 18.3% [19]. A study conducted in the USA revealed that women with hypertensive pregnancy disorders should have a comprehensive plan of care, which includes prenatal counselling, frequent visits during pregnancy, timely delivery, appropriate intrapartum monitoring and care, and postpartum follow-up [20].

The guidelines for the management of hypertension before, during, and after pregnancy state that treatment is mandatory for severe hypertension [21]. A study conducted in India revealed that, in developing countries, women have little or no information on the types of pregnancy-induced complications or where to go for complications [22]. A systematic review conducted in Ethiopia revealed that poor access to health care, distance from health facilities, and level of education are the main causes of maternal and infant mortality [23]. A study conducted in rural Tanzania showed that telecenters, libraries, and other health information providers' services are gender sensitive to facilitate equitable access to relevant information for women for rational decision-making[24]. In a study conducted in low-income countries, there were no differences according to socioeconomic status or ethnicity, but younger women were more likely to say that their mothers were their most useful source of information [25].

A study conducted in Sweden revealed that pregnant women are recommended to engage in physical activity for 30 minutes without any contradiction [26].

In the study conducted at the university hospital, there was a significant correlation between educational level and information seeking from the internet, medical books, and telephone helplines. Persons with a higher level of formal education use these sources more than do people with less education [27].

The purpose of this research was to measure the information-seeking behaviour of pregnant women with respect to PIH complications and the factors that lead to PIH complications. Additionally, this research aimed to identify the main source of information concerning PIH-related complications among pregnant women. In general, this research is used to identify the magnitude of PIH in the study area, which helps to reduce the prevalence of PIH complications and to identify the right information source on PIH complications, where to go when PIH complications occur and how to address PIH-related education in the study area.

## Methods

### Study design, setting, and period

A community-based cross-sectional study design was conducted in rural Sekela woreda pregnant women. Sekela is found 165 km south of Bahir Dar. Gish Abay is a town of Sekela Woreda, located 430 km from Addis Ababa, Ethiopia. The distance from Bahirdar to Addis *Ababa **is 349 km,*** and the road distance is 551.4 km. The 2007 national census was conducted by the Central Statistical Agency of Ethiopia (CSA). Sekela Woreda has a total population of 138,691, an increase of 61.36% over the 1994 census, of whom 69,018 were men and 69,673 were women; 6,779 or 4.89% were urban inhabitants (46). Socioeconomic status is based on farming and commerce. In Sekela Woreda, there are 30 rural Kebeles. Among these Kebele, there are 1413 total pregnant women in rural Sekela Woreda. This report was accessed from Sekela woreda health officer focal person.

This study was conducted from May 15 to June 15, 2022, in rural Sekela Woreda, northwest Ethiopia.

The source population was pregnant women who had lived in rural Sekela Woreda, and the study population was pregnant mothers in rural Sekela Woreda.

### Inclusion and exclusion criteria

The inclusion criterion was pregnant women who lived permanently in rural Sekela Woreda. Pregnant women who were admitted for delivery services and who temporarily visited the study area were excluded from the study.

### Sample size and sampling procedures

Owing to the lack of a prior study in a similar study setting, we used 50% prevalence to determine the sample size. The sample size for this study was calculated via a single population proportion formula.

$$n = \frac{\left(\frac{z}{2}\right)^2 * p(1-p)}{d2}$$ Where n= the required sample size. Z = the value of the standard normal distribution corresponding to α/2= 1.96. p = predicted proportion of PIH information-seeking behaviour of pregnant women. q = 1-p predicted the proportion of PIH information-seeking behaviour of pregnant women. d= margin of error 5% (0.05). By using a single population proportion formula, we took a proportion (p) of 50%.

$$n = \frac{\left(\frac{z}{2}\right)^2 * p(1-p)}{d2} \quad , n = \frac{(1.96)^2 * 0.5(1-0.5)}{(0.05)2} = (3.8416 * 0.25)/0.0025 = 384$$

Two sample sizes were calculated. The maximum sample size was 384 on the basis of the proportion of HISB among pregnant women. With a 10% nonresponse rate, the final sample size was (384*10%) +384 = 423. With the use of a single population proportion formula and a nonresponse rate of 10%, the sample size was 423. Owing to the sampling stage, we used Design Effect 1.5 and multiplied it by the sample size. The final calculated sample size was 635.

### Operational definition

Information seeking behaviour: PIH-Information Seeking behaviour is characterized by the PIH information sources, types of PIH information, and trust in PIH information sources [5,6,28]

PIH information seeking: We were measured through questioner. Whether pregnant women know PIH before or not. Yes or no question. The response is coded as 0 for no and 1 for yes correspondingly [5,29].

Use of information sources: - We asked participants how often (never, sometimes, often, or all the time) they used different information sources to receive pregnancy-induced hypertension health information in the past 12 months, then identify the most frequently used source for seeking information about PIH by using the mean score to identify their information source[5,30].

Alcohol drinking: The safest choice for a woman who is pregnant or planning to become pregnant is not to drink alcohol. Therefore, if she says drinks Alcohol, and if not, she is not drinking alcohol [31,32].

Distance: This is the distance from a pregnant woman to the health facility. It was measured as greater than or equal to 60 minutes thus far and less than 60 minutes not far [23,33].

Health efficacy: Measures how confident the pregnant women are about their aptitude to take good care of their health [5,28,29].

Perceived susceptibility to PIH: Assess the perception of the risk of contracting a PIH. We were measured as concerned and not concerned [5,29].

Perceived severity: Assess the seriousness of contracting a PIH. The respondent's responses were severe and the respondent's responses were not severe [29,34].

## Data collection tool and procedure

A pre-test and structured face-to-face interview were used to collect data from the study participants. The study's instrument (tool) was adapted from the health information national survey tool [35]. This was done to adapt the PIH information-seeking behaviour questionnaire. Different questions measuring sociodemographic characteristics, PIH information sources, behavioural characteristics, technological factors, and environmental factors were included in the data collection tool [8,11,28,29]. The data collection procedure was a face-to-face interview via an interviewer-administered Amharic questionnaire in the rural Sekela woreda community. The data were collected by nine data collectors that involved BSc health informaticians and BSc midwifery professionals. Two supervisors (a health informatician and a midwife professional) were recruited to follow up on the data collection procedure. The principal investigator controlled all the activities.

## Data quality assurance

One day of training was given for data collectors and supervisors on research ethics, providing informed consent, data collection procedures, data collection tools, how to approach participants, data confidentiality, and respondent rights, and all the study protocols were followed throughout the data collection period. Continuous monitoring was performed by supervisors throughout the data collection period to ensure that the data were collected according to the study protocol. Before the actual data collection, a pre-test was conducted among pregnant women. Before the study period at Megdom Aser Kebele, which is the Quarite district near Sekela woreda.

## Measurements

PHI information seeking was measured with one item derived from a previous study that determined whether respondents had sought PIH information in the past 12 months[30,36]. The participants were subsequently asked about the frequency of PIH information seeking. PIH information-seeking behaviour can be characterized by PIH information sources, types of PIH information, trust in PIH information sources, and reasons for seeking PIH information.

## Data processing and analysis

The data entry form was created via Epi-data version 4.6, and the analysis was carried out via STATA version 14.1. To identify factors associated with PIH-related information-seeking behaviour, bivariable and multivariable logistic regression were used. Descriptive statistics such as frequencies, means, and standard deviations were computed to summarize the study results. The binary logistic regression method was used to analyse the association between each independent variable and the dependent variable (PIH information-seeking behaviour) and to select candidate variables for the multiple logistic regression analyses. Variables with a p value of less than or equal to 0.2 in the bivariate logistic regression were entered into the multivariable logistic regression model. Finally, the multivariable logistic regression model was fitted to identify factors associated with participants' information-seeking behaviour related to PIH. The global model fitness of the final model was checked via the Hosmer and Lemeshow test, in which its P value was 0.9103, indicating that the model was a good fit, and a 95% confidence interval with a corresponding p value of less than 0.05 was used to declare statistical significance.

## Ethical approval

The study was conducted after ethical approval was obtained from the Institutional Review Board (IRB) of the University of Gondar. In addition to the IPH official permission letter, we obtained an official permission letter from the Sekela woreda health office, and informed written consent was also obtained from each study participant before the actual data collection started. The participants were informed about the risks and benefits of the study; they had the right to withdraw anytime, that confidentiality was maintained via codes, and that their right to obtain results for free.

## Results

### Sociodemographic characteristics

A total of 604 pregnant women participated in this study, with a response rate of 95%. Most participants (419; 69.4%) were farmers, 67 (11.09%) were merchants, 53 (8.77%) were employers, 46 (7.6%) were housewives, and the remaining 30.6% were included in other businesses, such as housewives, merchants, and 19 (3.15%) were private businesses. The mean age of the participants was 31.8 ± 6.09 years, with minimum and maximum ages of 20 and 45 years, respectively. More than half of the mothers 348 (57%) were < 35 years old. With respect to the marital status of the participants, 512 (84.77%) were married. More than half of the participants (354, 58.61%) could not read and write, but 41.39% of the participants could read and write. The demographic distribution of the participants is presented in (Table 1).

### Technological characteristics

The majority of the study participants, 384 (63.58%), did not use a mobile phone. Only 220 (36.4%) of the participants were mobile users. Twenty-five (11.36%) of the participants were current mobile internet users. The internet satisfaction level of 16 (64%) participants was very satisfied, and that of 6 (24%) participants was satisfied (Table 2). However, the majority of the study participants did not use mobile internet for PIH information-seeking purposes. Among the 25 study participants who used the internet, 10 (40%) were not used for PIH disease-related information-searching purposes (Table 2).

### Behavioural and health-related characteristics

Among the 604 participants, 392 (64.9%) were current alcohol drinkers. Most of the mothers drank Tela and Arekie 212 (53.94%) and 179 (45.55%), respectively. According to our findings,

**Table 1. Sociodemographic characteristics of pregnant women in Rural Sekela woreda (N = 604).**

| Variables | Category | Frequency | Percent (%) |
|---|---|---|---|
| Age | >=35 years old | 256 | 42.38 |
| | <=34 years old | 348 | 57.6 |
| Marital Statues | Married | 512 | 84.77 |
| | Single | 43 | 7.12 |
| | Divorced | 49 | 8.11 |
| Educational Statues | Cannot read & write | 354 | 58.61 |
| | Can read and write | 51 | 8.44 |
| | Primary schooling | 109 | 18.05 |
| | Secondary schooling | 37 | 6.13 |
| | Diploma and more | 53 | 8.77 |
| Maternal occupation | Farmer | 419 | 69.37 |
| | Private business | 19 | 3.15 |
| | Merchant | 67 | 11.09 |
| | Housewife | 46 | 7.62 |
| | Employee | 53 | 8.77 |

**Table 2. Technological factors of PIH information seeking for pregnant women in Rural Sekela Woreda North, Ethiopia, in 2022.**

| Variables | Category | Frequency | Percent (%) |
|---|---|---|---|
| Phone owner | Yes | 220 | 36.42 |
| | No | 384 | 63.58 |
| Internet access currently | Yes | 25 | 11.36 |
| | No | 195 | 88.64 |
| Level of Satisfaction with the internet | Very satisfied | 16 | 64.00 |
| | Satisfied | 6 | 24.00 |
| | Unsatisfied | 2 | 8.00 |
| | Very unsatisfied | 1 | 4.00 |
| Use of internet for PIH information | Yes | 15 | 60.00 |
| | No | 10 | 40.00 |

the majority of pregnant women drink Tela on occasion, up to two glasses, and the others drink Arekie and Tela up to three glasses or three melekiya of drunk alcohol per occasion. The results revealed that there were no cigarette smokers among the study participants (Table 3). Among the total participants, 393 (65.07%) were not confident in their health. Among the total sample, 211 (34.93%) of the respondents were confident in their health with respect to PIH disease prevention. Among the participants, 130 (21.52%) were resisted by their family to access PIH information by their husbands 63 (48.8%) and by their religious fathers 53 (41.09%). Among the study participants, 571 (94.54%) had no PIH signs or symptoms. However, most of the participants did not experience signs or symptoms, and 360 (59.6%) did not feel healthy (Table 3).

## Psychological characteristics

Among the total respondents, approximately 122 (20.20%) were concerned with having PIH for their lifetime, and 482 (79.8%) were not concerned with having PIH disease in their lifetime. However, most of the participants did not hesitate to suspect PIH. A total of 212 (35.1%) agreed that PIH complications are deadly diseases. On the other hand, 392 (64.9%) did not agree that PIH

**Table 3. Behavioural characteristics of PIH in pregnant women in Rural Sekela woreda in northern Ethiopia, 2022.**

| Variables | Category | Frequency | Percent (%) |
|---|---|---|---|
| Current drinker | Yes | 392 | 64.90 |
| | No | 212 | 35.10 |
| Types of drinking | Arekie/Katikal | 179 | 45.55 |
| | Tela | 212 | 53.94 |
| | Beer | 2 | 0.51 |
| Cigarette smoking | Yes | 0 | 0.00. |
| | No | 604 | 100.00 |
| Would you say your health is safe? | Confident | 211 | 34.93 |
| | Not confident | 393 | 65.07 |
| Family history of PIH | Yes | 194 | 32.12 |
| | No | 335 | 55.46 |
| | I'm not sure | 75 | 12.42 |
| PIH signs and symptoms | Yes | 33 | 5.46 |
| | No | 571 | 94.54 |
| Health statues | feel healthy | 244 | 40.4 |
| | Not feel healthy | 360 | 59.6 |
| Resistance of family | Yes | 130 | 21.52 |
| | No | 474 | 78.48 |
| The most protest of Mothers | My husband | 63 | 48.84 |
| | My children | 4 | 3.10 |
| | My husband's family | 4 | 3.10 |
| | My own family | 6 | 4.65 |
| | Religious Fathers | 53 | 41.09 |

is a deadly disease. Among the participants, only 208 (34.44%) perceived the severity of PIH. The results revealed that the participants were not psychologically ready for the occurrence of pregnancy-induced hypertension complications in their lifetime. This led to an increase in the prevalence of PIH disease in the study area. One in five (20.2%) pregnant mothers were concerned about the likelihood of having pregnancy-induced hypertension. Approximately one-third (34.4%) and 412 (65.6%) of pregnant women believe that PIH is severe and fatal, respectively (Table 4).

## Environmental characteristics

A total of 461 (76.32%) of the participants travelled to the health center after walking for more than one hour. A total of 143 (23.68%) patients took less than one hour to obtain health services from the health center. The results revealed that 461 (76.3%) of the participants had a health center problem, and when we looked at health care professionals' service satisfaction, 393 (65.07%) of the participants were not satisfied with the care provider's service, and 211 (34.93%) of the study participants were satisfied with the care provider's service. When we look over the advice accessed from healthcare providers, 34.6% of the pregnant mothers received advice from healthcare providers (Table 5).

## Nutritional characteristics

In total, 415 (69.28%) of the study participants did not use fruit. On the other hand, 184 (30.73%) used fruit in their diets. More than half of the participants did not use sugary foods

**Table 4. Psychological characteristics of pregnant women in Rural Sekela Woreda. North, Ethiopia 2022.**

| Variables | Category | Frequency | Percent (%) |
|---|---|---|---|
| Likely get PIH | Concerned | 122 | 20.20 |
| | Not concerned | 482 | 79.80 |
| Perceive the Severity of PIH | Severe | 208 | 34.44 |
| | Not sever | 396 | 65.56 |
| PIH is a deadly disease? | I Agree | 212 | 35.1 |
| | I'm not Agree | 392 | 64.90 |

**Table 5. Environmental factors for pregnant women with PIH disease in rural Sekela woreda north, Ethiopia, 2022.**

| Variables | Category | Frequency | Percent (%) |
|---|---|---|---|
| Distance of health centre from home | Far | 461 | 76.3 |
| | Not far | 143 | 23.68 |
| Health professional service satisfaction | Yes | 211 | 34.93 |
| | No | 393 | 65.07 |
| Care provider advice | Yes, I advise | 209 | 34.6 |
| | I did not advice | 395 | 65.4 |
| Health centre problem | Yes | 461 | 76.3 |
| | No | 143 | 23.68 |

*<60' means less than one hour, *>=60' means greater than one hour.

during pregnancy, as shown below (Table 6). A total of 486 (80.46%) no iodinated salt users were included, and the remaining 118 (19.54%) iodinated salt users were included. Among the total participants, 293 (48.8%) were eating vegetables. Among the total participants, 311 (51.49%) were not vegetarian. A total of 97.3% of the participants used salt all the time while cooking, and only 16 (2.65%) of the participants used salt sometimes while cooking (Table 6).

## Frequency of PIH Information seeking

Of the total PIH information sought by participants, 140 (65.47%) participants sought PIH information once a month, and 64 (30.14%) of the participants sought PIH information occasionally. The other was sought by 7 (3.39%), 2 (0.79%), and 1 (0.19%) individual every day, within three months, and once every six months, respectively (Fig 1).

## For whom the participants sought PIH information

Among the total information seeker participants, 190 (88.9%) sought PIH information for themselves, 20 (9.4%) sought PIH information for their family and friends, and the remaining 4 (1.68%) were for family and self (Fig 2).

## Source of PIH health information for pregnant women

The participants were asked about their sources of health information for PIH disease. As shown in the figure below, 124 (58.1%) participants used healthcare providers as their PIH information source, and 67 (31.29%) participants used family or friends as a source of PIH information. Therefore, we can say that books, newspapers, radio, and the internet had fewer users for PIH information as a source (Fig 3).

**Table 6. Nutritional characteristics of pregnant women in rural Sekela woreda, North Ethiopia.**

| Variables | Category | Frequency | Percent (%) |
|---|---|---|---|
| Sugary foods | Yes | 282 | 46.69 |
| | No | 322 | 53.31 |
| Usage of fruit per week | I did not eat fruit | 415 | 69.28 |
| | I eat fruit | 184 | 30.73 |
| Types of salt usage | Iodinated | 118 | 19.54 |
| | Non-Iodinated | 486 | 80.46 |
| Frequency of salt usage | Always | 588 | 97.35 |
| | Sometimes | 16 | 2.65 |
| Usage of vegetable per week | Yes | 293 | 48.51 |
| | No | 311 | 51.49 |

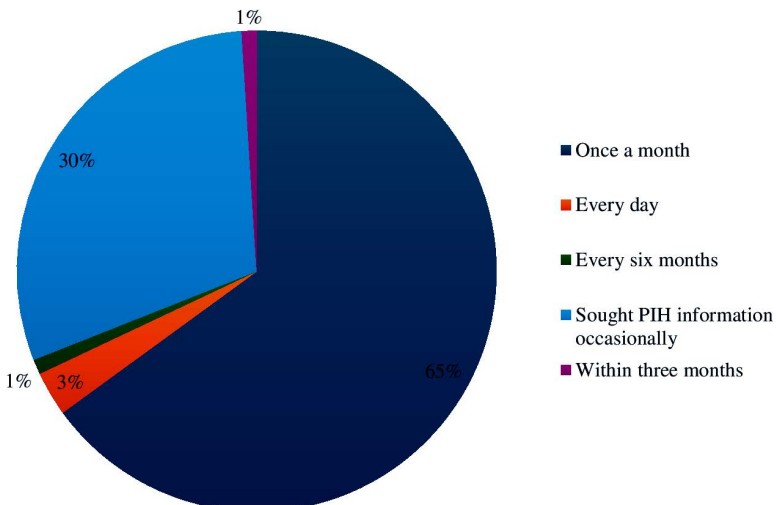

**Fig 1. Frequency of seeking PIH information for pregnant women in rural Sekela woreda north, Ethiopia 2022.**

## Type of PIH Information

Among the total information sought by the participants, 131 (61.3%) sought information for all types of PIH. The other participants were for specific PIH disease types for which they sought PIH information. As shown in the graph below, 39 (18.1%), 31 (14.5%), and 13 (6%) participates sought preeclampsia, gestation, and eclampsia PIH information.

## Information sources trusted by pregnant women for PIH

Among the study participants, 110 (51.6%) trusted healthcare providers, and 79 (37.1%) trusted their family or friends for their PIH information source. The remaining 12 (5.64%), 7 (3.35%), 4 (1.37%), and 2 (0.91%) trusted the internet, radio, newspapers, and books, respectively (Fig 4).

## Reason for not seeking information on PIH

Among the total participants who had not sought PIH information, 268 (68.79%) were afraid of seeking PIH information. Ninety (23.3%) of the participants also believed that seeking PIH

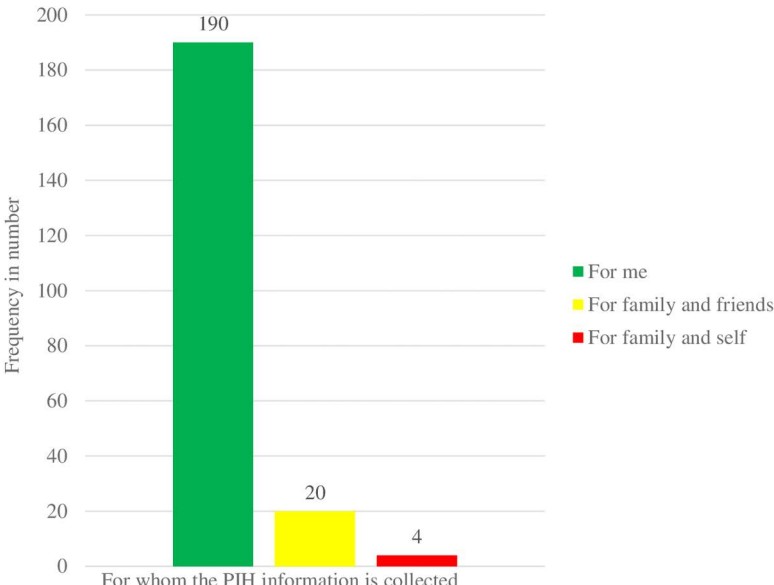

**Fig 2. For whom pregnant women sought PIH information rural Sekela woreda north, Ethiopia, 2022.**

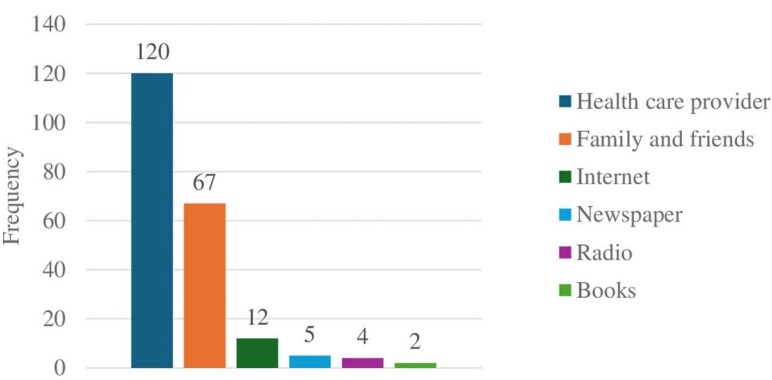

**Fig 3. Source of health information in PIH disease for pregnant women in rural Sekela woreda north, Ethiopia, 2022.**

information with other people, including care providers, was inappropriate, while 25 (6.2%) believed that PIH information was uninteresting, and the remaining 7 believed that online information was from an untrusted source(Table 7).

Proportion of information seeking related to pregnancy-induced hypertension. Overall, 214 (35.4%) participants sought PIH information (Fig 5).

## Factors associated with PIH information-seeking behaviour

Binary and multivariable logistic regression was used to select significant factors associated with PIH information-seeking behaviour. The results of the bivariable logistic regression analysis included the following: sociodemographic characteristics (age), technological characteristics (phone owner), psychological characteristics (self-efficacy and perceived severity of PIH),

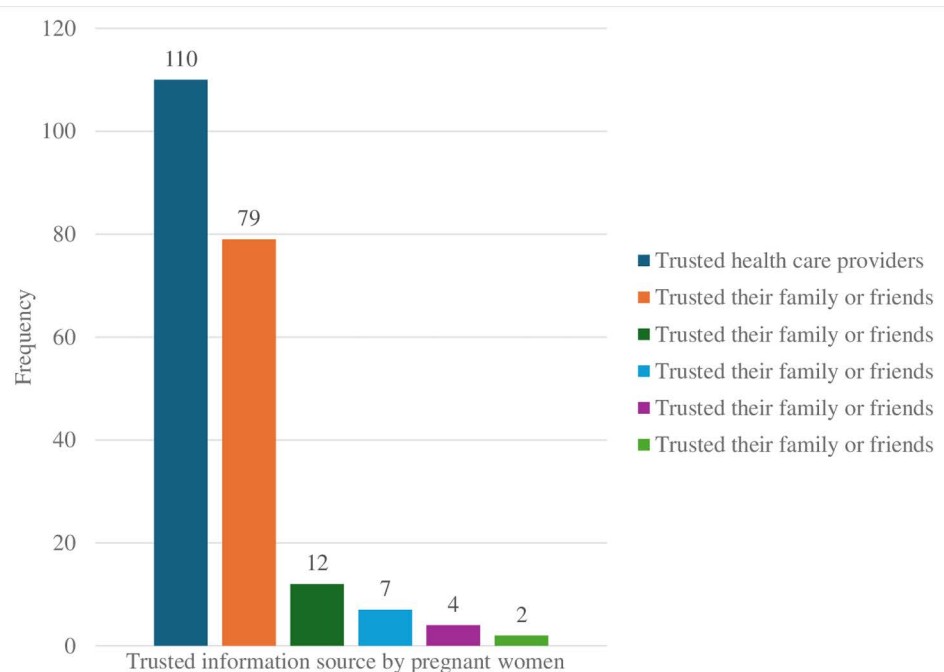

**Fig 4. Trusted PIH information Source for pregnant women in rural Sekela woreda north, Ethiopia, 2022.**

**Table 7. Reasons for not seeking PIH information in rural Sekela woreda north, Ethiopia, 2022.**

| Reasons | Frequency | Percent (%) |
|---|---|---|
| Fear about becoming difficult with PIH | 268 | 68.79 |
| PIH information is perceived as irrelevant | 90 | 23.32 |
| PIH information is not interesting | 25 | 6.21 |
| Mistrust of online information | 7 | 1.68 |

environmental/organizational characteristics (service satisfaction and distance), health-related characteristics (death and ANC follow-up), and behavioural characteristics (fruit consumption and family resistance), which were significantly associated with PIH information-seeking variables, with P values less than 0.2. In a multivariable analysis, the mother's age, family resistance, perceived severity of PIH, and health care service satisfaction were significantly associated with PIH information-seeking behaviour, with P values less than 0.05 for discussion. The odds of information seeking among pregnant mothers aged>= 35 years were 33% (AOR = 0.67, 95% CI =.29,.69) less likely to seek PIH-related information than those of women aged under 35 years. The family resistance of pregnant women is another significant variable for the outcome variable. Therefore, pregnant women who were resisted by their family (AOR= 0.45, 95% CI =.26,.63) were 55% less likely to seek PIH information than pregnant women who were not resisted by their family to seek PIH information. With respect to health service satisfaction, pregnant women who were satisfied with health care services were 1.7 (AOR =.59, 95% CI = 1.1, 2.5) times more likely to seek PIH information than pregnant women who were not satisfied with the health service provided by health care providers. In addition, the severity of PIH was another significant factor for the outcome variable. Pregnant women who knew

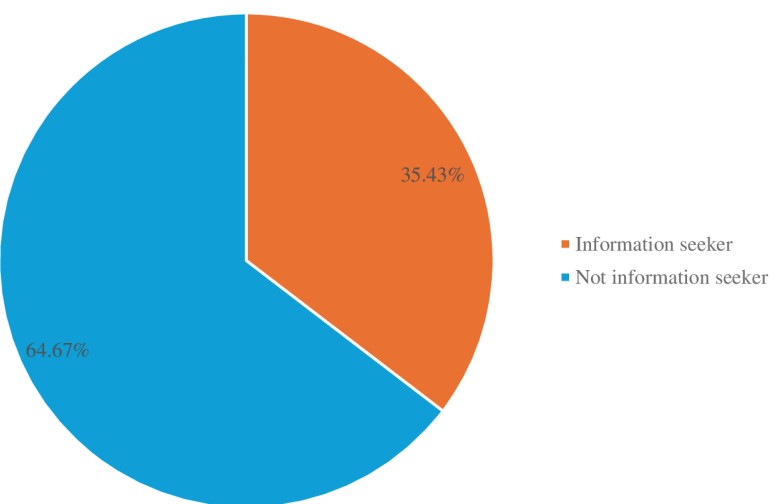

**Fig 5. Proportion of information seeking related to PIH.**

the severity of PIH were 1.6 (AOR = 1.6, 95% CI = 1.1, 2.4) times more likely to seek PIH information than participants who did not know the severity of PIH. The other variables were not significantly associated (Table 8)

## Discussion

This study examined the information-seeking behaviours of pregnant women regarding pregnancy-induced hypertension (PIH) in rural Sekela woreda, along with the challenges they face. The findings will help assess the feasibility of desired information sources on PIH and identify related factors. This research aims to inform policymakers and stakeholders in developing strategies to reduce the burden of PIH.

According to the findings of this study, 35.4% (95% CI = 31%, 39%) of pregnant women sought PIH information from various information sources in the previous 12 months, which is lower than that reported in studies conducted in the Tigray region [19]. The possible reason might be due to differences in the residences of the study participants. The study was conducted in Tigray in a hospital, but the current study was conducted in a rural area. Pregnant women in rural Ghana sought PIH information from various sources, but in the current study, healthcare providers were the primary source, compared to previous studies that used family and friends. Language barriers, geographical location, and technology inaccessibility also contributed to the challenges in finding information seekers. [37]. With respect to PIH information sources, the current study indicates that 58.1% of pregnant women used healthcare professionals as their first PIH information source. This study is lower than the study conducted in selected hospitals in the Ibadan Metropolis; the former study revealed that 93.4% of the respondents' PIH information sources were health care professionals. This difference might be due to the participants' study area and educational status. In the former study, the participants were urban pregnant women, but the current study was conducted in rural areas [38]. In the current study, 51.6% of the respondents trusted that their information sources were healthcare providers. Similarly, in a study conducted at Florida State University, the trusted information sources of pregnant women were health professionals [39]. The findings of the study conducted at Florida State University concerning the source of PIH information are consistent with those of the current study, in which 58.1% of the participants

**Table 8. Factors associated with PIH information seeking among pregnant women in Sekela Woreda, North Ethiopia, 2022.**

| Information sought to recode | PIH information-seeking | | COR (95%) | AOR (95%) |
|---|---|---|---|---|
| | Yes N (%) | No N (%) | | |
| Age of the mother | | | | |
| < 35 years | 137(39.4) | 211(60.6) | 1 | 1 |
| >=35 years | 77(30) | 179(69.9) | 0.66(0.47 - 0.93) | 0.67 (0.46 - 0.97) * |
| Resistance of family for information of PIH | | | | |
| No | 152(32.1) | 322 (67.9) | 1 | 1 |
| Yes | 62(47.6) | 68(52.3) | 0.51(0.34 - 0.76) | 0. 45 (0.29 – 0.69) ** |
| Phone owner | | | | |
| No | 125(34.9) | 258(66.1) | 1 | 1 |
| Yes | 89(41.5) | 131(58.5) | 1.37(1.1 - 1.9) | 1.23(.89 - 1.8) |
| Self-efficacy | | | | |
| Confident | 90(42.05) | 124(57.9) | 1 | 1 |
| Not confident | 124(31.8) | 266(68.2) | 0.68(0.48 – 0.96) | 0.47 (0.69 - 1.00) |
| Distance of health centre | | | | |
| Not far | 58(40.5) | 85(59.4) | 1 | 1 |
| Far | 156(33.8) | 305(66.2) | 0.74(0.50 - 0.99) | 0.89(.56 - 1.3) |
| ANC follow-up | | | | |
| No | 76(32) | 161(67.9) | 1 | 1 |
| Yes | 138(37.6) | 229(62.3) | 1.2(0.90 - 1.8) | 0.95(.64 - 1.4) |
| Perceive the severity of PIH | | | | |
| Not sever | 55(26.4) | 153(73.5) | | 1 |
| Sever | 159(40.1) | 237(59.9) | 1.8(1.2 - 2.6) | 1.6(1.1 - 2.4) * |
| Is PIH a deadly disease? | | | | |
| Agree | 91(42.9) | 121(57.1) | 1 | 1 |
| Disagree | 123(31.4) | 269(68.6) | 0.60(.43 - 0.85) | 0.70(.48 - 1.01) |
| Satisfaction of care provider service | | | | |
| No | 120(30.6) | 273(69.4) | 1 | 1 |
| Yes | 94(44.5) | 117(55.6) | 1.8(1.2 - 2.5) | 1.7(1.1 - 2.5) * |

Reference

*= p value <0.05, ** p value =0.000, and 1 = reference category

used healthcare providers as their PIH information source [39]. A study conducted in rural Malaysia revealed that 58.8% of the participants used their family or friends as their primary source of health information. This value is greater than that in the current study (32.2%). This difference might be due to family literacy and financial factors prohibiting the use of other sources, such as smartphones and magazines [8].

We discovered that pregnant women over the age of 35 were less likely to seek PIH information than were those under the age of 35. A study conducted in South Korea revealed that aged women had less intention to seek health information than young women did [40–42]. The possible reason might be that the aged women were not able to access different sources of information, and those women were leaving their time with their children, cooking, and other home activities. Furthermore, compared with young women, older women are less literate. A study conducted in the southern provinces of Laos reported that aged women did not actively communicate and accept the innovation, which led to fewer information seekers than younger women [16].

The findings of the study conducted in Northwest Health centres of Tehran contradict those of the former study, as age is not a significant factor for PIH information seeking [43]. The possible reason might be the educational level of the study participants. Almost all were diplomas and above. In addition, the former study participants were urban, but the current study participants were rural farmers [43]. According to a study conducted in China, older pregnant mothers were more comfortable discussing pregnancy-related complications than younger or less than 35-year-old mothers were. However, in our study, most of the mothers could not read or write, which contradicts our findings; however, younger women were more likely to seek pregnancy-related complications than older mothers were [42].

Family resistance was found to be a significant factor related to pregnant women's health information seeking. The study revealed that 21.65% of the pregnant women, particularly their husbands, faced resistance from their families. The majority of these women reported that their families did not utilize medical treatment [13]. The current study is similar to a study conducted in India, which reported that women faced opposition from their families, especially their husbands' families, if they refused to perform household tasks, not only for the sake of time but also because they were not interested in seeking information from health facilities [22]. The qualitative study conducted in India revealed that women seek different information from their husbands and mothers, which is due to the shying of mothers to ask professionals and because "my mother is also a woman and she knows pregnancy best. Therefore, why do we need to go to the doctor for advice" [11].

The study, which was conducted in low-income countries, revealed that pregnant women were aware of healthy behaviours and reported practicing them during their pregnancies. Family members are a common source of information for health practices, they should be included in health education efforts. In the present study, families resisted pregnant mothers from health facilities, and families were not given health education for PIH because of their lack of awareness of health education. The current findings were not similar to those of the former study because the former study's family was aware of health education as well as diagnosis, but in our study, the family was not aware of health education and diagnosis for the complication or disease of PIH[44].

A qualitative study conducted in Ethiopia revealed that the health care-seeking behaviour of pregnant women was influenced during pregnancy, birth, and postpartum by religious and cultural factors[45] Therefore, the former study similarly stated that family resistance was a factor for pregnancy-induced hypertension disorders compared with the current findings. A study conducted at the University of Dodoma, Tanzania, reported that when a pregnant woman wanted to seek health information related to her pregnancy, the family resisted seeking any pregnancy-related information from health facilities. Instead, they use Traditional Birth Attendance and mothers-in-law. This is due to a lack of healthcare professionals in Tanzania [46].

The perceived severity of pregnancy-induced hypertension (PIH) is another significant factor related to PIH information seeking. Participants who perceived the severity of PIH were 1.6 times more likely to seek PIH information than those who did not perceive the severity [47]. A study conducted at the University of Gondar reported that the participants had high perceived severity, and they had a high intention of knowing the risk factors, prevention methods, diagnosis, and treatment of the disease[5]. Our findings are similar to those of a study conducted in the Dawro zone, Esera woreda, which reported that perceived severity is significantly associated with health care-seeking information. Therefore, these severely ill participants were more likely to seek healthcare information than those the participants who were not severely ill[48]. A study conducted in Esera woreda, Ethiopia, reported that severely ill participants were more information seekers than those who were not severely ill [48].

Health care satisfaction is another significant factor for the outcome variable. Our results show that the participants who were satisfied with the service of the health care provider were more likely to be PIH information seekers than the participants who were not satisfied with the service. This finding is similar to that of a study conducted in Sweden, which reported that most of the participants who were satisfied with the health care provider service were more likely to seek information than the participants who were not satisfied with the service of the health care provider[27]. The current findings are supported by a study conducted in Kano Metropolis, Nigeria, which reported that participants who were satisfied with the antenatal care service in the health facility were more information seekers than participants who were not satisfied with the antenatal care service in the facility [49].

## Strengths and limitations of the study

Most of the studies conducted on information seeking in different cases were hospital- and organizational-based, but this study was conducted in a rural community setting, which is difficult in terms of material feasibility and ease of task accomplishment. As I have seen in different studies, this finding was the first finding in a rural area. Therefore, these findings might be used as a reference for future researchers.

The limitation of this research is that it does not include all rural pregnant women in our sample because of time and resource limitations. The study was a community-based cross-sectional study, which may not articulate causal inference between variables.

## Conclusion and recommendations

The study revealed a significant level of pregnancy-induced hypertension (PIH) information-seeking behaviour among pregnant women. The main sources of PIH information were healthcare professionals and families, but healthcare providers were the most trusted source. Among the information-seeking participants, most sought details on various PIH complications and for themselves. These participants demonstrated awareness of managing and preventing PIH complications. The study revealed several factors significantly associated with PIH information-seeking behaviour, including age, family resistance, satisfaction with care providers, and perceived severity of PIH. These findings suggest the need to expand health education programs and increase awareness of PIH among pregnant women.

Training participants and their husbands in PIH is the most effective way to reduce the prevalence of PIH complications.

## Supporting information

**S1 Data. Raw data for pregnant women.**
(DTA)

## Acknowledgement

We would like to express our deepest gratitude to the University of Gondar, the Sekela woreda health office, the study participants, the data collectors, and the Department of Health Informatics.

## Author contributions

**Conceptualization:** Ayana Muluneh Alebachew, Fekade Demeke Bayou, Kegnie Shitu, Ayenew Sisay Gebeyew, Sefefe Birhanu Tizie, Mulugeta Desalegn Kasaye, Adamu Ambachew Shibabaw, Agmasie Damitew Walle.

**Data curation:** Ayana Muluneh Alebachew, Fekade Demeke Bayou, Ayenew Sisay Gebeyew, Sefefe Birhanu Tizie, Mulugeta Desalegn Kasaye, Agmasie Damitew Walle.

**Formal analysis:** Ayana Muluneh Alebachew, Ayenew Sisay Gebeyew, Sefefe Birhanu Tizie, Agmasie Damitew Walle.

**Investigation:** Ayana Muluneh Alebachew.

**Methodology:** Ayana Muluneh Alebachew.

**Project administration:** Ayana Muluneh Alebachew.

**Resources:** Ayana Muluneh Alebachew.

**Software:** Ayana Muluneh Alebachew.

**Supervision:** Kegnie Shitu, Ayenew Sisay Gebeyew.

**Validation:** Ayana Muluneh Alebachew.

**Visualization:** Ayana Muluneh Alebachew, Kegnie Shitu.

**Writing – original draft:** Ayana Muluneh Alebachew.

**Writing – review & editing:** Ayana Muluneh Alebachew.

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
