## [Decision Letter · Decision Letter 0]

7 Aug 2024

PDIG-D-24-00137

Maternal Information-Seeking on Pregnancy-Induced Hypertension and Associated Factors among Pregnant Women, in Low resource country, A cross sectional study design

PLOS Digital Health

Dear Dr. Alebachew,

Thank you for submitting your manuscript to PLOS Digital Health. After careful consideration, we feel that it has merit but does not fully meet PLOS Digital Health's publication criteria as it currently stands. Therefore, we invite you to submit a revised version of the manuscript that addresses the points raised during the review process.

Please submit your revised manuscript within 60 days Oct 06 2024 11:59PM. If you will need more time than this to complete your revisions, please reply to this message or contact the journal office at digitalhealth@plos.org. Please include the following items when submitting your revised manuscript:

We look forward to receiving your revised manuscript.

Kind regards,

Jasmit Shah, PhD

Guest Editor

PLOS Digital Health

Journal Requirements:

Additional Editor Comments (if provided):

Overall, the paper is not yet fully aligned with the publication standards for PLOS Digital Health. The use of English is below average and requires to meet international standards.

The research lacks a clear emphasis on its novelty and unique contributions.

Need to use relevant guidelines (e.g., CONSORT, STROBE).

Reviewers' comments:

Reviewer's Responses to Questions

**Comments to the Author**

1. Does this manuscript meet PLOS Digital Health’s publication criteria ? Is the manuscript technically sound, and do the data support the conclusions? The manuscript must describe methodologically and ethically rigorous research with conclusions that are appropriately drawn based on the data presented.

Reviewer #1: Partly

Reviewer #2: Partly

Reviewer #3: No

2. Has the statistical analysis been performed appropriately and rigorously?

Reviewer #1: I don't know

Reviewer #2: Yes

Reviewer #3: No

3. Have the authors made all data underlying the findings in their manuscript fully available (please refer to the Data Availability Statement at the start of the manuscript PDF file)?

Reviewer #1: Yes

Reviewer #2: No

Reviewer #3: No

4. Is the manuscript presented in an intelligible fashion and written in standard English?

PLOS Digital Health does not copyedit accepted manuscripts, so the language in submitted articles must be clear, correct, and unambiguous. Any typographical or grammatical errors should be corrected at revision, so please note any specific errors here.

Reviewer #1: No

Reviewer #2: No

Reviewer #3: No

5. Review Comments to the Author

Please use the space provided to explain your answers to the questions above. You may also include additional comments for the author, including concerns about dual publication, research ethics, or publication ethics. (Please upload your review as an attachment if it exceeds 20,000 characters)

Reviewer #1: Thank you for your nice work, the topic is interesting, and this study can add value to the concept of perinatal health.

I would like to mention to some point you need to address:

• In the result section of the abstract, line 37, you mentioned that the proportion of PIH information seeking among pregnant women was 214?? 214 out of what, either right the total number or keep only the percentage.

• Typo in line 80.

• I think the sentence in lines 89, 90 is in complete, what is the relationship??

• Typo in line 130.

• Where you interviewed or you did interview pregnant women, line 152.

• Review line 167.

• Typo in 210

These are examples or errors in your paper.

I recommend that you revise your manuscript and submitted again for review.

Reviewer #2: The paper explores the impact of digital health interventions on managing Pregnancy-Induced Hypertension (PIH) in rural Ethiopia. To strengthen its scientific impact, consider addressing the following aspects:

1. Originality and Novelty:

o Clearly highlight the novel aspects of your research and how it differs from existing studies. A comparative analysis with recent similar research can help establish the unique contributions of your work.

2. Importance and Broad Interest:

o Emphasize the broader implications of your findings. Articulate how your study influences practice, policy, or further research in digital health, and how it could be relevant beyond the specific context of rural Ethiopia.

3. Methodological Rigor and Ethical Standards:

o Provide a more detailed description of your study design, data collection methods, and analysis techniques. Ensure all ethical considerations are thoroughly addressed, including obtaining necessary approvals.

4. Substantial Evidence:

o Include additional analyses or data to further substantiate your conclusions. Address any study limitations openly and discuss how future research could address these limitations.

5. Utility and Accessibility:

o Elaborate on the practical applications of your findings. Ensure that the implications for practice and policy are clear and accessible to a wide audience, including practitioners and policymakers.

6. Open Science Standards:

o Deposit your data and related protocols in public repositories to enhance transparency and reproducibility. Provide links or access information in your manuscript.

7. Reproducibility and Protocols:

o Detail your methodological protocols to facilitate replication of your study. If there were deviations from standard practices, explain and justify these deviations clearly.

8. Clear and Accessible Writing:

o Revise the manuscript to improve clarity and coherence. Ensure that complex concepts are explained in an accessible manner for non-specialists.

9. Supporting Information:

o Consider including detailed protocols, algorithms, or additional data as supplementary materials. This can enhance the transparency and depth of your research.

10. Compliance with Guidelines:

o Confirm and mention adherence to relevant guidelines (e.g., CONSORT, STROBE) to ensure the methodological rigor of your study.

Overall, the paper is not yet fully aligned with the publication standards for PLOS Digital Health. The research lacks a clear emphasis on its novelty and unique contributions, making it difficult to discern how it differentiates itself from existing studies in the field. The evidence provided to support the conclusions is insufficient, as the analysis and data are not comprehensive enough to firmly substantiate the claims made. The methods section needs greater detail to ensure reproducibility, which is a crucial aspect of scientific rigor.

Ethical considerations are underrepresented, with insufficient discussion of approvals and ethical guidelines pertinent to the study. Additionally, the practical utility of the findings is not well articulated, leaving the potential applications and implications unclear. The manuscript also falls short in adhering to open science practices, as there is a lack of publicly accessible data and detailed protocols.

Furthermore, the clarity of the writing requires improvement to ensure the manuscript is accessible to a broader, non-specialist audience. Essential supporting materials, such as detailed protocols or algorithms, are missing, which hinders the transparency of the research. Lastly, the study does not reference adherence to relevant research guidelines like CONSORT or STROBE, impacting its methodological rigor and alignment with established research standards.

Reviewer #3: The use of English is below average and requires professional proof reading to meet international standard.

The scope of this study seems not to align with digital health in anyway, I would suggest the author consider submission to pure public health or health communication journal

6. PLOS authors have the option to publish the peer review history of their article (what does this mean? ). If published, this will include your full peer review and any attached files.

**Do you want your identity to be public for this peer review?** For information about this choice, including consent withdrawal, please see our Privacy Policy .

Reviewer #1: Yes: Nisreen Al Jallad

Reviewer #2: No

Reviewer #3: No

---

## [Decision Letter · Decision Letter 1]

19 Nov 2024

PDIG-D-24-00137R1Maternal Information-Seeking on Pregnancy-Induced Hypertension and Associated Factors among Pregnant Women, in Low resource country, A cross sectional study design

 PLOS Digital Health

Dear Dr. Alebachew,

Thank you for submitting your manuscript to PLOS Digital Health. After careful consideration, we feel that it has merit but does not fully meet PLOS Digital Health's publication criteria as it currently stands. Therefore, we invite you to submit a revised version of the manuscript that addresses the points raised during the review process.

Please submit your revised manuscript within 30 days Dec 19 2024 11:59PM. If you will need more time than this to complete your revisions, please reply to this message or contact the journal office at digitalhealth@plos.org. Please include the following items when submitting your revised manuscript:

* A rebuttal letter that responds to each point raised by the editor and reviewer(s). You should upload this letter as a separate file labeled 'Response to Reviewers '. This file does not need to include responses to any formatting updates and technical items listed in the 'Journal Requirements' section below.

* A marked-up copy of your manuscript that highlights changes made to the original version. You should upload this as a separate file labeled 'Revised Manuscript with Track Changes '.

* An unmarked version of your revised paper without tracked changes. You should upload this as a separate file labeled 'Manuscript '.

We look forward to receiving your revised manuscript.

Kind regards,

Jasmit Shah, PhD

Guest Editor

PLOS Digital Health

Jasmit Shah

Guest Editor

PLOS Digital Health

Leo Anthony Celi

Editor-in-Chief

PLOS Digital Health

orcid.org/0000-0001-6712-6626

**Additional Editor Comments (if provided):**

**Reviewers' Comments:**

Reviewer's Responses to Questions

**Comments to the Author**

1. If the authors have adequately addressed your comments raised in a previous round of review and you feel that this manuscript is now acceptable for publication, you may indicate that here to bypass the “Comments to the Author” section, enter your conflict of interest statement in the “Confidential to Editor” section, and submit your "Accept" recommendation.

Reviewer #1: All comments have been addressed

Reviewer #4: (No Response)

2. Does this manuscript meet PLOS Digital Health’s publication criteria ? Is the manuscript technically sound, and do the data support the conclusions? The manuscript must describe methodologically and ethically rigorous research with conclusions that are appropriately drawn based on the data presented.

Reviewer #1: Yes

Reviewer #4: (No Response)

3. Has the statistical analysis been performed appropriately and rigorously?

Reviewer #1: I don't know

Reviewer #4: No

4. Have the authors made all data underlying the findings in their manuscript fully available (please refer to the Data Availability Statement at the start of the manuscript PDF file)?

Reviewer #1: Yes

Reviewer #4: Yes

5. Is the manuscript presented in an intelligible fashion and written in standard English?

Reviewer #1: Yes

Reviewer #4: No

6. Review Comments to the Author

Reviewer #1: It is nicely done.

Reviewer #4: (No Response)

7. PLOS authors have the option to publish the peer review history of their article (what does this mean? ). If published, this will include your full peer review and any attached files.

**Do you want your identity to be public for this peer review?** For information about this choice, including consent withdrawal, please see our Privacy Policy .

Reviewer #1: **Yes: ** Nisreen Al Jallad

Reviewer #4: No

**Figure resubmission:**
---

## [Editor Report · Decision Letter 2]

6 Jan 2025

Maternal Information-Seeking on Pregnancy-Induced Hypertension and Associated Factors among Pregnant Women, in Low resource country, A cross sectional study design

PDIG-D-24-00137R2

Dear Alebachew,

We are pleased to inform you that your manuscript 'Maternal Information-Seeking on Pregnancy-Induced Hypertension and Associated Factors among Pregnant Women, in Low resource country, A cross sectional study design' has been provisionally accepted for publication in PLOS Digital Health.

Best regards,

Jasmit Shah, PhD

Guest Editor

PLOS Digital Health